# *Acinetobacter baumannii* Sampled from Cattle and Pigs Represent Novel Clones

Valeria Mateo-Estrada,[a] Leila Vali,[b] Ahmed Hamouda,[c] Benjamin A. Evans,[d] (ID) Santiago Castillo-Ramírez[a]

[a]Programa de Genómica Evolutiva, Centro de Ciencias Genómicas, Universidad Nacional Autónoma de México, Cuernavaca, México
[b]Medical Laboratory Sciences, Faculty of Allied Health, Kuwait University, Sulaibekhat, Kuwait
[c]Birmingham, United Kingdom
[d]Norwich Medical School, University of East Anglia, Norwich, United Kingdom

**ABSTRACT** *Acinetobacter baumannii* is a very important human pathogen. Nonetheless, we know very little about nonhuman isolates of *A. baumannii*. Here, we determine the genomic identity of 15 Scottish cattle and pig isolates, as well as their antibiotic and virulence genetic determinants, and compare them with 148 genomes from the main human clinical international clones. Our results demonstrate that cattle and pig isolates represent novel clones well separated from the major international clones. Furthermore, these new clones showed fewer antibiotic resistance genes and may have fewer virulence genes than human clinical isolates.

**IMPORTANCE** Over the last decades, huge amounts of information have been obtained for clinical isolates of *A. baumannii* and the clones they belong to. In contrast, very little is known about the genomic identity and the genomic basis for virulence and resistance of animal isolates. To fulfil this gap, we conducted a genomic epidemiology study of 15 Scottish cattle and pig isolates in the context of almost 150 genomes belonging to the main international clones of *A. baumannii*. Our findings show that these animal isolates represent novel clones clearly different from the major international clones. Furthermore, these new clones are distinct in nature considering both antibiotic resistance and virulence when compared with their human clinical counterparts.

**KEYWORDS** genome epidemiology, *Acinetobacter baumannii*, antibiotic resistance, animal isolates, bacterial clones, One Health

*A*cinetobacter baumannii is a Gram-negative opportunistic bacterial pathogen, notorious for being associated with high morbidity and mortality due to its highly drug-resistant nature. While *A. baumannii* can be isolated from clinical samples, its natural environment is less clear. Animals have been suggested as a potential host or reservoir for *A. baumannii*. Birds, in particular White Storks, have been proposed as a reservoir (1), though this does not seem to apply to other bird species (2), and as *A. baumannii* can be released into the environment from hospital effluent (3) it is not clear the degree to which wild animals are acquiring the bacteria from contaminated soil and water. However, it is clear that *A. baumannii* should be considered a One Health issue, as some nonhuman isolates have important antibiotic resistance genes (4). *A. baumannii* seems to be fairly common in domestic livestock, particularly cattle (5, 6), where isolates tend to have a generally susceptible antibiotic resistance profile and appear to be genetically distinct from clinical strains by molecular typing methods. In a previous study, 16 *A. baumannii* isolates were collected from cattle and pigs that had been recently slaughtered, and were shown by pulsed-field gel electrophoresis (PFGE) to cluster separately from the three major clones of *A. baumannii* prevalent at the time; furthermore, they carried different *oxaAb* (*bla*OXA-51-like) variants (7). Here, we sequenced the genomes of these 16 isolates to determine how genetically similar they are to human clinical isolates.

**Ad Hoc Peer Reviewer** (ID) Mattia Pirolo, Università Roma Tre

Address correspondence to Santiago Castillo-Ramírez, iago@ccg.unam.mx, or Benjamin A. Evans, Benjamin.evans@uea.ac.uk.
The authors declare no conflict of interest.

Total DNA was extracted from overnight broth cultures with a Promega Wizard Genomic DNA Purification kit (Promega, UK), quality checked by nanodrop and quantity assessed by Qubit. Purified DNA was paired-end sequenced on an Illumina platform. The sequences were trimmed with Trim Galore (8) and assembled via SPAdes (9), as described previously (10). The genomes were annotated employing Prokka (11) and genotyped by the Pasteur Multilocus Sequencing Typing (MLST) scheme (12) using the PubMLST online database (13). The genome quality was assessed with CheckM (14) and only the genomes with more than 95% completeness and less than 5% contamination were considered for downstream analyses. One isolate from a pig fecal sample (PF33) was discarded as it showed a high percentage of contamination (>60%). For the phylogenetic analysis, we also included 148 human-related *A. baumannii* genomes previously genotyped in Hernández-González et al. (15). These genomes were chosen as they are part of the eight main international clones (ICs). Table S1 provides the BioSample ID for all the isolates and also some other information such as host, isolation source, geographic location, ST assignation, etc. A maximum likelihood (ML) core phylogeny was built using the strategy described in Graña-Miraglia (16). Briefly, the genes present in a single copy in all the genomes (single-gene families) were identified with Roary (17) and tested for recombination using PhiPack (18). We found 759 single-gene families without recombination, which represent 47.8% of the core genome, and these were concatenated to build a phylogeny with RAxML (19), the tree was annotated using iTOL (20). The antibiotic resistance genes prediction on the genome assemblies was carried out with the Comprehensive Antibiotic Resistance Database (CARD) (21), and *ampC* alleles were identified using the PubMLST database.

The ML core genome phylogeny of the 15 animal isolates alongside a collection of 148 clinical isolates (22) representing the major international clones showed that the animal isolates formed three well-separated clades, each of which was distinct and very distant from any of the clinical isolates (Fig. 1). The pig fecal isolates formed a single clade, two of the cattle fecal isolates (CF233 and CF234) formed a second clade, and the remaining four cattle fecal isolates formed a clade with the two cattle nostril isolates. Considering the Pasteur MLST genotyping, these three clades corresponded to sequence type (ST$^{PAS}$)162, ST$^{PAS}$1014, and ST$^{PAS}$492, respectively. As described previously, the isolates belonging to ST$^{PAS}$1014 carried the *oxaAb* variant *oxaAb*(150), and the ST$^{PAS}$492 isolates carried *oxaAb*(148) (7). However, the ST$^{PAS}$162 strains carried *oxaAb*(51), which is considered diagnostic of international clone (IC) 4 isolates (23). IC4 isolates typically belong to ST$^{PAS}$15, which only shares a single allele in common with ST$^{PAS}$162, and the ST$^{PAS}$162 and IC4 isolates are very clearly separated in the core genome phylogeny (Fig. 1). It is interesting to note that incongruence between MLST ST and *oxaAb* allele has been observed previously for ST$^{PAS}$162, and warrants further investigation (24). Of note is that all published ST$^{PAS}$162 isolates, and all of those in the PubMLST database, are from South American countries (Brazil and Chile), geographically very distant from the Scottish isolates reported here. Only two ST$^{PAS}$492 isolates are listed in the PubMLST database, from Lebanon and Russia, and one of these is an animal isolate. There are no other ST$^{PAS}$1014 isolates in the PubMLST database, but there are 11 isolates that match six loci, and of these, two are listed as coming from animals, two from food, and one from the environment, suggesting that these STs are commonly isolated from nonclinical sources. Collectively, these results show that the pig and cattle isolates form well-differentiated groups and they are not closely related to the major international clones.

As expected from the previously reported generally antibiotic sensitive nature of the isolates, only chromosomally encoded resistance genes such as *oxaAb*, *ampC*, and efflux systems were identified, with no acquired antibiotic resistance genes present (Fig. 2). Of note, we identified novel ampC alleles in the isolates: ampC-84 was present in CF233 and CF234; ampC-85 was found in CN26, CN35, CF251, CF254, CF258, and CF260; and ampC-86 was present in the rest of the isolates. It had previously been described for these animal isolates that they did not carry IS*Aba1* upstream of the *oxaAb* genes or the *ampC* genes, where it can provide a promoter for their expression

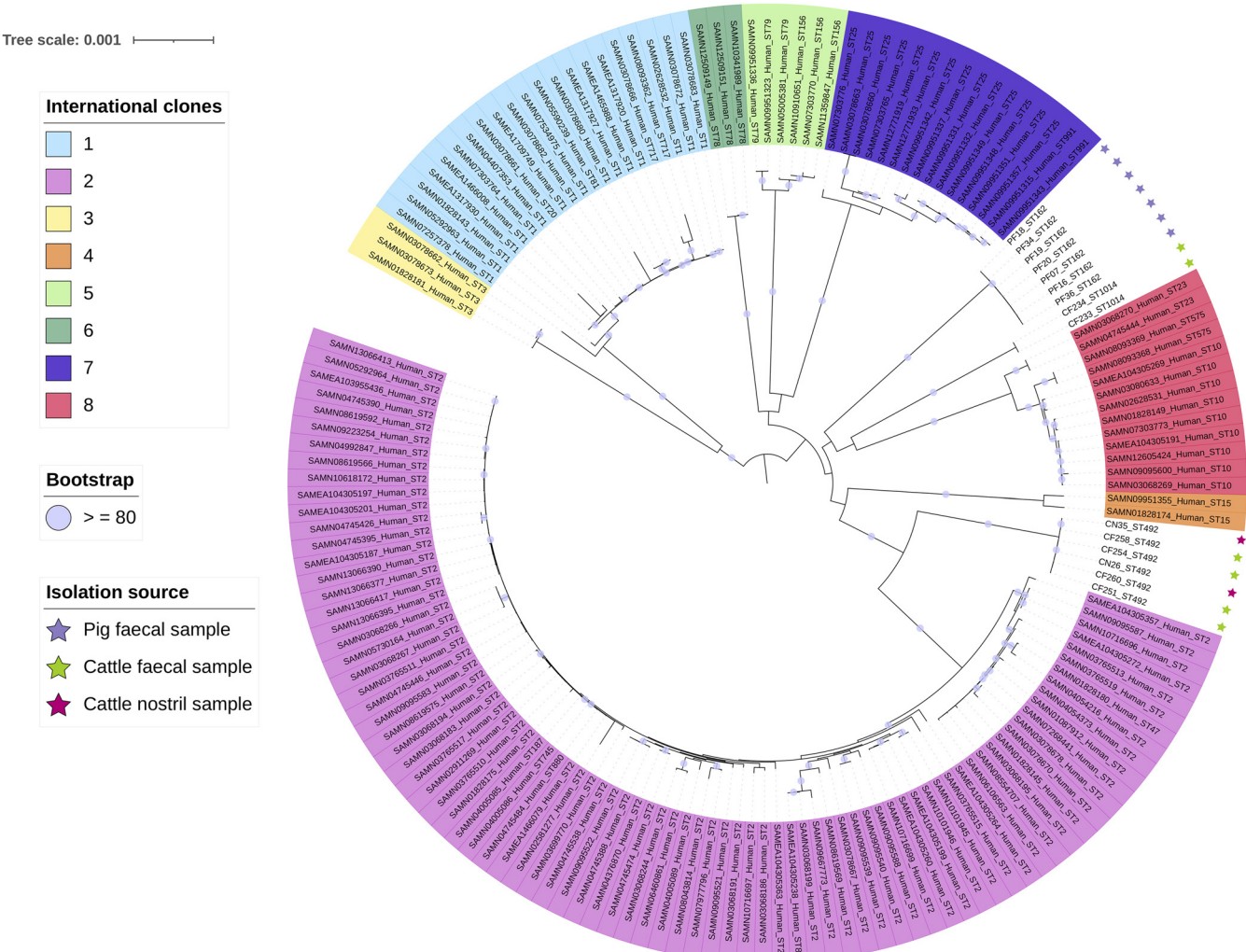

**FIG 1** Core genome maximum likelihood phylogeny of animal isolates and 148 clinical isolates from eight international clonal lineages. The major international clones are highlighted with different colors on the labels. Pig isolates are denoted with violet stars, whereas cattle isolates are shown with green (fecal samples) and rosy (nostril samples) stars. The tree scale is the number of substitutions per site and bootstrap values higher (or equal to) 80 are depicted with violet circles at the internal nodes of the phylogeny.

(7). Of the 15 isolates, seven had no substantial match to any insertion sequences in the ISFinder database (accessed 09/03/2022). Of the remaining eight, one isolate (CN26) had a short contig with a partial match to IS*1411*, while all seven pig fecal isolates contained a 184-bp fragment with 97% identity to the 5' end of IS*Ajo2*, and a 279 bp fragment with 84% similarity to the 3' end of IS*Acsp2*. We, therefore, did not detect any complete IS elements in these strains. In *A. baumannii*, IS elements are thought to be a major mechanism through which the bacteria regulate gene expression and mobilize genes, and are a common feature of clinical isolates. Their almost complete absence from these animal isolates highlights how different in nature they are from clinical isolates, and that IS-mediated adaptation may be a feature of successful clinical strains rather than a general characteristic of the species.

In order to assess whether the animal isolates differed in their complement of virulence factors, the genome assemblies were analyzed using VFanalyzer alongside all 15 available genomes included in the VFanalyzer database (14 clinical and one human louse isolate; Table S2) (25). These genomes represent six different MLST$^{PAS}$ STs, including eight ST$^{PAS}$2 and three ST$^{PAS}$1 genomes. Animal isolates had a significantly smaller complement of capsule-related genes, averaging 17, whereas the clinical isolates averaged 22 (t test, P = 0.000016). The six ST$^{PAS}$492 isolates differed from the other animal

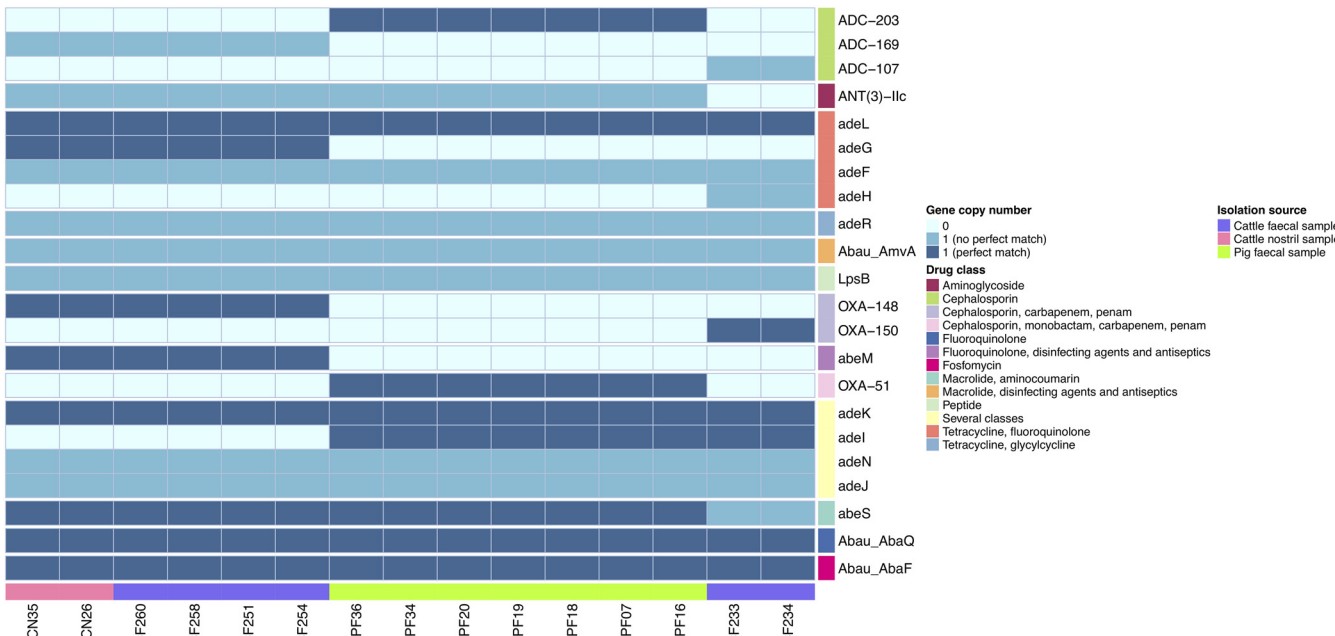

**FIG 2** Matrix showing the presence/absence of genes involved in antibiotic resistance in the animal isolates. The genes are color-coded according to the drug classes they confer resistance to. Dark blue denotes perfect matches (100% identical) to alleles already described in CARD, whereas the sky blue shows significant but not perfect matches (not 100% identical); light blue shows cases where no match was found.

isolates in that they lacked 8 genes involved in heme utilization, including *hemO* (Table S2). Variation in the carriage of these genes is common, with seven out of the 14 clinical and one louse strains also lacking these genes. Thus, these data suggest that the animal isolates may have fewer virulence factors than human clinical isolates.

In conclusion, our study shows that these cattle and pig isolates represent three novel clones well-separated from the major international clones. Furthermore, these new clones are distinct in nature considering both antibiotic resistance and virulence when compared with their human clinical counterparts. In a broader context, our findings highlight the need for further studies on the genomic epidemiology, and also surveillance of animals isolates of this bacterial species.

**Data availability.** The animal isolates were submitted to the NCBI under the BioProject number PRJNA819013. In addition, the BioSample number for each isolate is listed in Table S1.

## SUPPLEMENTAL MATERIAL

Supplemental material is available online only.
**SUPPLEMENTAL FILE 1**, XLSX file, 0.02 MB.
**SUPPLEMENTAL FILE 2**, XLS file, 0.1 MB.

## ACKNOWLEDGMENTS

We are thankful to Ana Mateus from MRCVS, Clinical Scholar, Animal Production and Public Health Department, Glasgow University Veterinary School for supplying the abattoir samples. We are also grateful to Alfredo José Hernández Álvarez and Victor Manuel del Moral Chávez for helping to install several of the programs employed in this work. In addition, we are appreciative of Joel Gómez Espíndola, Iván Uhthoff Aguilera, and Maria Gabriela Guerrero Ruiz for technical support on diverse aspects.

This work was supported by CONACyT Ciencia Básica 2016 (grant no. 284276) and "Programa de Apoyo a Proyectos de Investigación e Innovación Tecnológica PAPIIT" (grant no. IN206019) given to S.C.-R. V.M.-E. is a doctoral student from the

Programa de Doctorado en Ciencias Biomédicas, Universidad Nacional Autónoma de México, and she is funded by a CONACYT doctoral fellowship (no. 1005234).

We declare no conflicts of interest.

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
