## [Reviewer comments · Microbiology Spectrum]

Microbiology Spectrum

***Acinetobacter baumannii* sampled from cattle and pigs represent novel clones**

Valeria Mateo-Estrada, Leila Vali, Ahmed Hamouda, Benjamin Evans, and Santiago Castillo-Ramírez

Corresponding Author(s): Santiago Castillo-Ramírez, Programa de Genómica Evolutiva, Centro de Ciencias Genómicas, Universidad Nacional Autónoma de México

Review Timeline:

Submission Date:	April 6, 2022
Editorial Decision:	May 3, 2022
Revision Received:	June 8, 2022
Accepted:	June 8, 2022

Editor: Paolo Visca

Reviewer(s): Disclosure of reviewer identity is with reference to reviewer comments included in decision letter(s). The following individuals involved in review of your submission have agreed to reveal their identity: Mattia Pirolo (Reviewer #2)

Transaction Report:

DOI: <https://doi.org/10.1128/spectrum.01289-22>

May 3, 2022

Dr. Santiago Castillo-Ramírez
Programa de Genómica Evolutiva, Centro de Ciencias Genómicas, Universidad Nacional Autónoma de México
Evolutionary Genomics Research Program
CCG UNAM
Cuernavaca 62210
Mexico

Re: Spectrum01289-22 (Acinetobacter baumannii sampled from cattle and pigs represent novel clones)

Dear Dr. Santiago Castillo-Ramírez:

Thank you for submitting your manuscript to Microbiology Spectrum.

Both Reviewers provided constructive criticism to your manuscript which needs to be carefully addressed: I am confident that your manuscript will become acceptable after revision.

Link Not Available

Sincerely,

Paolo Visca

Journals Department
Reviewer comments:

Reviewer #2 (Public repository details (Required)):

Whole-genome sequencing data for the 15 *Acinetobacter baumannii* isolates should be submitted to a publicly available repository of high throughput sequencing data (i.e., NCBI SRA or ENA).

Reviewer #2 (Comments for the Author):

In the paper entitled "*Acinetobacter baumannii* sampled from cattle and pigs represent novel clones", Authors sequenced and analysed *A. baumannii* isolates collected from Scottish cattle and pigs. While the rationale of studying the genetic background of

these isolates is interesting, a main criticism to the Manuscript is that sequencing is limited to only 16 isolates, of which one was discarded from analysis due to contamination. The 15 isolates were genetically compared to a collection of 148 human-related *A. baumannii* isolates. Results highlighted that animal isolates i) formed separate clusters genetically distant from human isolates, and ii) showed a reduced number of antimicrobial resistance and virulence genes compared to human isolates. I have some minor suggestions to improve the quality of the manuscript.

1. Please submit whole-genome sequencing data for the 15 *A. baumannii* isolates to a publicly available repository of high throughput sequencing data (i.e., NCBI SRA or ENA).
2. Please consider changing the title to "*Acinetobacter baumannii* sampled from Scottish cattle and pigs represent novel clones" to better represent the main focus of the study
3. Lines 32, 43-44, 150-152. Please change the term "lineages" with "clones".
4. Line 45. Please spell "antibiotic resistance and virulence genes".
5. Lines 33-34. Please change to "these new clones showed less antibiotic resistance and virulence genes".
6. Line 40. Please change "to alleviate this" with "to fulfil this gap".
7. Line 55. Please clarify what "natural home" refers to or rephrase.
8. Line 66. It is unclear from which site the 16 strains were isolated. Please specify.
9. Line 83. Please provide additional information on the 148 *A. baumannii* isolates selected for comparison, especially isolation sites as "human-related" is too generic. Data can be provided as supplementary material.
10. Lines 86-88. Which is the size of pan and core genome? Please express the 759 concatenated genes as proportion of core genome.
11. Line 91. It is unclear if assembled genomes or raw reads were screened against CARD. Please specify. The same applies to the virulence factor analysis.
12. Lines 139-141. Please specify how the 15 isolates from humans were selected for virulence factor analysis, especially because this selection could have biased the following statistical analysis.
13. Lines 150-152. Please change the term "lineages" with "clones".
14. Reference 15 is incomplete. Please revise.
15. Figure 2 legend. How distances between isolates (top tree) and genes (side tree) were calculated? Please explain what the dashed line on the side tree refers to.
16. Please revise Affiliation no. 3, which at present is "Birmingham, UK".

Reviewer #3 (Public repository details (Required)):

The genome sequencing data need to be deposited and accession provided.

Reviewer #3 (Comments for the Author):

The study is experimentally/methodologically solid. I only have two points of criticism:

1. The genome sequencing data need to be deposited and accession provided.
2. The in silico analysis of virulence factors does not justify statements such as "These new lineages have less...virulence..." in the abstract. Rather, it is as written at the end of the manuscript that "...these data suggest that the animal isolates may have fewer virulence factors than human clinical isolates."

Staff Comments:

Preparing Revision Guidelines

Please return the manuscript within 60 days; if you cannot complete the modification within this time period, please contact me. If you do not wish to modify the manuscript and prefer to submit it to another journal, please notify me of your decision immediately so that the manuscript may be formally withdrawn from consideration by Microbiology Spectrum.

Reviewer comments:

Reply: We sincerely thank both reviewers for the time invested in reviewing our manuscript and their constructive comments. The lines mentioned in the answers refer to the Mark-Up copy of the manuscript.

Reviewer #2 (Public repository details (Required)):

Whole-genome sequencing data for the 15 *Acinetobacter baumannii* isolates should be submitted to a publicly available repository of high throughput sequencing data (i.e., NCBI SRA or ENA).

Reply: We thank both reviewers for noticing this. We have submitted the 16 isolates to the NCBI. The BioProject number has been included in the "Data availability" section at the end of the manuscript and the BioSample ID for each isolate have been included in Supplementary Table 1.

Reviewer #2 (Comments for the Author):

In the paper entitled "*Acinetobacter baumannii* sampled from cattle and pigs represent novel clones", Authors sequenced and analysed *A. baumannii* isolates collected from Scottish cattle and pigs. While the rationale of studying the genetic background of these isolates is interesting, a main criticism to the Manuscript is that sequencing is limited to only 16 isolates, of which one was discarded from analysis due to contamination. The 15 isolates were genetically compared to a collection of 148 human-related *A. baumannii* isolates. Results highlighted that animal isolates i) formed separate clusters genetically distant from human isolates, and ii) showed a reduced number of antimicrobial resistance and virulence genes compared to human isolates.

I have some minor suggestions to improve the quality of the manuscript.

Reply: We do thank the reviewer for taking the time to carefully review our work. Her/his comments about our work have been included in the revised version of the manuscript.

1. Please submit whole-genome sequencing data for the 15 *A. baumannii* isolates to a publicly available repository of high throughput sequencing data (i.e., NCBI SRA or ENA).

Reply: We thank both reviewers for noticing this. We have submitted the 16 isolates to the NCBI. The BioProject number has been included in the "Data availability" section at the end of the manuscript and the BioSample ID for each isolate have been included in Supplementary Table 1.

2. Please consider changing the title to "*Acinetobacter baumannii* sampled from Scottish cattle and pigs represent novel clones" to better represent the main focus of the study

Reply: Thanks for the suggestion. We prefer to leave the title as it stands to emphasize the general trend. However, we have slightly edited the text to clearly state in the abstract and the importance section that the isolates were sampled in Scotland.

3. Lines 32, 43-44, 150-152. Please change the term "lineages" with "clones".

Reply: Thanks for the suggestion. We have replaced "lineages" with "clones"

4. Line 45. Please spell "antibiotic resistance and virulence genes".

Reply: Done (see lines 35 and 36)

5. Lines 33-34. Please change to "these new clones showed less antibiotic resistance and virulence genes".

Reply: Done (see line 35)

6. Line 40. Please change "to alleviate this" with "to fulfil this gap".

Reply: This has been changed (see line 42)

7. Line 55. Please clarify what "natural home" refers to or rephrase.

Reply: Thanks for noticing this. We have rephrased that part - now it reads "natural environment" (see line 65)

8. Line 66. It is unclear from which site the 16 strains were isolated. Please specify.

Reply: Thanks for noticing this. We have added a new Supplementary Table (Supplementary Table 1, in the revised version) where the source of isolation is stated for each isolate.

9. Line 83. Please provide additional information on the 148 *A. baumannii* isolates selected for comparison, especially isolation sites as "human-related" is too generic. Data can be provided as supplementary material.

Reply: Thanks for the suggestion. A new Supplementary Table was added (see comment above also) that lists the metadata for the isolates. There for each isolates you can find the BioSample ID, source of isolation, geographic origin, ST assignation, isolate date, etc.

10. Lines 86-88. Which is the size of pan and core genome? Please express the 759 concatenated genes as proportion of core genome.

Reply: The size of the pangenome is 15,374 gene families and the core genome is 1588 gene families. The 759 non-recombinant families represent 47.8% of the core genome; this last point has been added to the manuscript (see line 105).

11. Line 91. It is unclear if assembled genomes or raw reads were screened against CARD. Please specify. The same applies to the virulence factor analysis.

Reply: Thanks for noticing this. This has been clarified.

12. Lines 139-141. Please specify how the 15 isolates from humans were selected for virulence factor analysis, especially because this selection could have biased the following statistical analysis.

Reply: These 15 isolates represent the total available in the VFanalyser tool. They represent 6 different MLST STs, with a bias towards ST1 (3 genomes) and ST2 (8 genomes), the most widespread clinical STs. Text to this effect has been added to the manuscript (lines 173 - 175).

13. Lines 150-152. Please change the term "lineages" with "clones".

Reply: Thanks for the suggestion. We have replaced "lineages" with "clones"

14. Reference 15 is incomplete. Please revise.

Reply: Thanks for noticing this. We have revised reference 15.

15. Figure 2 legend. How distances between isolates (top tree) and genes (side tree) were calculated? Please explain what the dashed line on the side tree refers to.

Reply: Thanks for the suggestion. For clarity sake, we have simplified the figure, taking out the clusterings on the top and the right of the figure. The clusterings were according to gene presence and absence (right) and strains (top) but they are not relevant to the points that we make in the manuscript.

16. Please revise Affiliation no. 3, which at present is "Birmingham, UK".

Reply: This author is not currently employed and as such does not have an official affiliation. We have therefore provided the location of the city where the author currently lives. We would welcome any guidance from the editorial team as to how to approach this unusual situation.

Reviewer #3 (Public repository details (Required)):

The genome sequencing data need to be deposited and accession provided.

Reply: We do thank the reviewer for her/his positives comments about our work. We did pay attention to include her/his comments in the new version of the manuscript.

Reviewer #3 (Comments for the Author):

The study is experimentally/methodologically solid. I only have two points of criticism:

Reply: We do thank the reviewer for her/his positives comments about our work. We did pay attention to include her/his comments in the new version of the manuscript.

1. The genome sequencing data need to be deposited and accession provided.

Reply: We thank both reviewers for noticing this. We have submitted the 16 isolates to the NCBI. The BioProject number has been included in the "Data availability" section at the end of the manuscript and the BioSample ID for each isolate has been included in Supplementary Table 1.

2. The in silico analysis of virulence factors does not justify statements such as "These new lineages have less...virulence..." in the abstract. Rather, it is as written at the end of the manuscript that "...these data suggest that the animal isolates may have fewer virulence factors than human clinical isolates."

Reply: Thanks for the suggestion. We have changed the statements about the virulence factors in the abstract as suggested by the reviewer (see line 35)

June 8, 2022

Dr. Santiago Castillo-Ramírez
Programa de Genómica Evolutiva, Centro de Ciencias Genómicas, Universidad Nacional Autónoma de México
Evolutionary Genomics Research Program
CCG UNAM
Cuernavaca 62210
Mexico

Re: Spectrum01289-22R1 (*Acinetobacter baumannii* sampled from cattle and pigs represent novel clones)

Dear Dr. Santiago Castillo-Ramírez:

Thank you for submitting the revised version of the manuscript "Acinetobacter baumannii sampled from cattle and pigs represent novel clones" to Microbiology Spectrum. I appreciate you addressed all comments of the two Reviewers and I believe that your manuscript has been substantially improved after revision.

Your manuscript has now been accepted, and I am forwarding it to the ASM Journals Department for publication. You will be notified when your proofs are ready to be viewed.

Sincerely,

Paolo Visca
Editor, Microbiology Spectrum

Journals Department
Supplemental Table 2: Accept
Supplemental Table 1: Accept